# Precision Medicine in Cholangiocarcinoma: Past, Present, and Future

**DOI:** 10.3390/life12060829

**Published:** 2022-06-02

**Authors:** Chi-Yuan Cheng, Chiao-Ping Chen, Chiao-En Wu

**Affiliations:** 1Department of Pharmacy, Chang Gung Memorial Hospital at Linkou, Taoyuan 333, Taiwan; cycdavid@cgmh.org.tw; 2Division of Hematology-Oncology, Department of Internal Medicine, Chang Gung Memorial Hospital at Linkou, Taoyuan 333, Taiwan; d000017242@cgu.edu.tw

**Keywords:** cholangiocarcinoma (CCA), targeted therapy, NTRK, IDH1, FGFR2, BRAF

## Abstract

Cholangiocarcinoma (CCA), or biliary tract cancer, has a poor prognosis. The median survival time among patients with CCA is under 2 years from diagnosis, and the global 5-year survival rate is only 10%. First-line therapy with chemotherapeutic agents, gemcitabine plus cisplatin, has traditionally been used to treat unresectable advanced CCA. In recent years, precision medicine has become a mainstream cancer treatment due to innovative next-generation sequencing technology. Several genetic alterations, including mutations, gene fusions, and copy number variations, have been found in CCA. In this review, we summarized the current understanding of genetic profiling in CCA and targeted therapy in CCA. Owing to the high heterogeneity of CCA, tumor microenvironmental factors, and the complexity of tumor biology, only pemigatinib, infigratinib, ivosidenib, larotrbctinib, and entrectinib are currently approved for the treatment of CCA patients with fibroblast growth factor receptor 2 gene (FGFR2) fusion, isocitrate dehydrogenase gene (IDH1) mutation, and neurotrophin receptor tyrosine kinase gene (NRTK) fusion, respectively. Additional targeted therapies, including other FGFR2 inhibitors, PI3K/AKT/mTOR inhibitors, and BRAF-directed targeted therapy, have been discussed for the management of CCA, and immune checkpoint inhibitors, particularly pembrolizumab, can be administered to patients with high microsatellite instability tumors. There is a further need for improvement in precision medicine therapies in the treatment of CCA and discuss the approved and potential targeted therapies for CCA.

## 1. Introduction

Cholangiocarcinoma (CCA), or biliary tract cancer (BTC), is a malignancy arising from the epithelium of the bile ducts [1]. Chronic inflammation in parasitic infections, primary sclerosing cholangitis, congenital fibropolycystic liver, hepatitis B and C viruses, and Caroli’s diseases are major causes of CCA [2,3]. Based on its anatomical origin, CCA can be either intrahepatic (iCCA) or extrahepatic (eCCA). The latter can be subdivided into perihilar (pCCA) and distal (dCCA). iCCA, pCCA, and dCCA account for 5–10%, 60–70%, and 20–30% of CCA cases, respectively [4]. In addition to CCA, both gallbladder and ampullary cancers are considered subtypes of BTC. This review focuses on CCA as a distinct genetic alteration between CCA and other malignancies.

CCA has a poor prognosis, as most cases are diagnosed at an advanced stage and respond poorly to current systemic treatment [1]. The CCA patients have medium survival of fewer than two years. In total, 90% of patients do not live more than five years after the first diagnosis [4]. Moreover, although the early stage of CCA can be completely removed by surgical resection, most patients experience recurrence within 2 years [5,6]. In terms of advanced or recurrent CCA, the median survival time is under 1 year [4,7,8], showing that it is a highly aggressive cancer that requires urgent attention.

First-line therapy with gemcitabine plus cisplatin, two chemotherapeutic agents, has traditionally been used for unresectable advanced CCA, despite the low efficacy of this approach based on the ABC-02 trial in 2010 [4,7,8]. Titanium silicate (TS-1) is another commonly used cytotoxic agent in Asia [9] and TS-1-based combinations, such as gemcitabine plus TS-1, have been widely studied [10,11]. Nevertheless, the lack of breakthroughs using these agents has given rise to the era of targeted therapy and immunotherapy. In recent years, the application of next-generation sequencing (NGS) technology has improved our knowledge of the molecular biology of cancer, as well as genetic alterations, and has expedited developments in targeted therapy for these alterations. Precision medicine involves the selection of suitable drugs, therapies, or preventive strategies based on an individual’s genotype or gene expression profile and clinical data, thereby achieving maximum therapeutic efficacy and minimal side effects [12,13,14]. This study describes the current state of precision medicine for CCA treatment.

## 2. Molecular Biological Classification of CCA

CCA can be divided into four clusters based on the genetic alterations and clinical features, in particular, whether it is associated with flukes and whether it is internal or external (Figure 1) [7]. Mutations in *IDH1*, *IDH2*, *BAP1*, and rearrangements in *FGFR2* are often present in iCCA, whereas *HER2* and *TP53* mutations are frequent in eCCA. Among the above genetic alterations, FGFR2 fusions may be associated with the best prognosis [7]. Although both iCCA and eCCA originate from the bile ducts, the distinct etiology may contribute to the different genetic alterations in CCA. The acronyms and full names of the genes are listed in Table 1.

In addition, the distinct genetic landscape of iCCA was found in Western and Asian populations [15]. Patients with a higher burden of DNA repair mutations and frequency consistent with high tumor mutational burden (TMB-H) have been reported in the Asian population than among Western patients [15]. This finding suggests the need for different therapeutic strategies, such as immunotherapy, in different populations.

## 3. Genetic Alterations in CCA and Clinical Trials of Targeted Therapies

Genetic alterations in CCA, including mutations, gene fusions, and copy number variations, can disrupt DNA repair, cell cycle regulation, signal transduction of receptor tyrosine kinases, and epigenetic regulation. Table 2 shows a list of genetic alterations in CCA and corresponding clinical trials [16,17,18]. Some of the genetic alterations currently have no clinical features; therefore, we provided potential targeted agents based on preclinical studies that are yet to be validated in human studies (Figure 2).

### 3.1. Fibroblast Growth Factor Receptor 2 Gene Fusions

Fibroblast growth factor receptors (FGFR, including FGFR1-5) comprise a family of receptor tyrosine kinases that regulate cell proliferation, differentiation, and migration upon the stimulation of fibroblast growth factor (FGF) [19]. *FGFR* gene alterations include indel mutations, amplifications, and fusions, leading to gain-of-function, which may drive CCA progression. Several small-molecule targeted therapies that block the FGF/FGFR signaling axis by tyrosine kinase inhibitors (TKIs) have been studied in CCA with *FGFR* alterations [20] and only CCA with *FGFR2* fusion can benefit from FGFR inhibitors. FGFR2 gene fusions, such as *FGFR2*–*BICCI*, *FGFR2*–*AHCYL1*, *FGFR2*–*TACC3*, and *FGFR2*–*KIAA1598*, are closely associated with tumor progression in iCCA [7]. The outcomes of clinical trials of small-molecule targeted therapies for these genetic alterations are presented in Table 3 [7,13]. Previous results indicated a maximum disease control rate (DCR) of 80% and progression-free survival (PFS) of approximately 6 months [14,20,21]; therefore, FGFR inhibitors are promising drugs for targeted therapies for CCA with FGFR2 fusions. The U.S. Food and Drug Administration (FDA) approved the use of pemigatinib and infigratinib for treating CCA patients with *FGFR2* gene fusions [22]. In addition, other FGFR inhibitors are under investigation in clinical trials.

### 3.2. Isocitrate Dehydrogenase 1 and 2 Mutations

Isocitrate dehydrogenase (IDH) is an enzyme involved in intracellular glucose metabolism. *IDH1* and *IDH2* mutations frequently occur in various myeloid malignancies and solid tumors [21]. Neomorphic mutation of IDH proteins produces the oncometabolite D-2-hydroxyglutarate, which blocks cellular differentiation via the inhibition of α-ketoglutarate-dependent dioxygenases involved in histone and DNA demethylation [23]; therefore, *IDH* mutations can result in abnormal cellular glucose metabolism, leading to DNA hypermethylation, abnormal cell proliferation, and angiogenesis [24]. *IDH* mutations are seemingly exclusive to iCAA rather than to eCCA. The results of a phase I study on CCA patients with IDH1 mutations treated with the IDH1 inhibitor ivosidenib (AG-120) revealed that the objective response rate (ORR) was 5%, median PFS was 3.8 months, and median overall survival (OS) time was 13.8 months [25]. In phase III, double-blind, randomized ClarIDHy clinical trial, ivosidenib was administered to patients with *IDH1* mutations, while a placebo was administered to those in the control group. The results showed that the PFS of the ivosidenib group was 2.7 months while that of the placebo group was 1.4 months (hazard ratio 0.37, *p* < 0.001). The ivosidenib group had an ORR of 2.4% and a DCR of 53.2%, whereas the placebo group had an ORR of 0% and a DCR of 27.9% [26], demonstrating the clinical efficacy of ivosidenib. Currently, it is approved for use in patients with *IDH1* mutations in acute myeloid leukemia (AML) [27] and CCA [28]. Ivosidenib in combination with nivolumab is under investigation for IDH1 mutant solid cancers (NCT04056910).

In terms of *IDH2* mutation, there are ongoing clinical trials on the application of the IDH2 inhibitor enasidenib (AG-221) (NCT02273739) and the pan-IDH inhibitor vorasidenib (AG-881), which inhibit IDH1 and IDH2 simultaneously in the treatment of CCA [24].

With the success of ivosidenib, targeting IDH mutations became possible; however, the activity of ivosidenib is limited, with a low ORR of 2.4%. The PFS benefit from ivosidenib largely results from a DCR of 53.2%; therefore, enhancing the activity of such inhibitors by modifying the structure or combination therapy should be a goal in patients with *IDH*-mutant CC.

### 3.3. Neurotrophic Tyrosine Receptor Kinase

Genomic translocation of the neurotrophic tyrosine receptor kinase gene (NTRK), which leads to the constitutive activation of receptor tyrosine kinases, is rare in pan-cancer, including CCA [29]; however, both larotrectinib and entrectinib have been approved for the treatment of cancers harboring NTRK gene rearrangements. Two of the fifty-five patients with various TRK fusion-positive malignancies were enrolled in three trials of larotrectinib treatment [30]. The overall ORR was 75%, and one out of two CCA cases had objective tumor shrinkage. Larotrectinib was approved by the FDA for use in adults and children with solid tumors with NTRK gene fusion in November 2018. Another drug, entrectinib, was approved by the FDA for cancers with an NTRK gene fusion in August 2019.

### 3.4. Overexpression of EGFR and HER2

Although epidermal growth factor receptor (EGFR) overexpression is rare, human epidermal growth factor receptor 2 (HER2) overexpression is common in CCA (particularly eCCA) and has a prevalence of 11–20% [4,18]. In terms of targeting EGFR, phase III studies on the use of the EGFR inhibitor erlotinib plus gemcitabine and cisplatin revealed that the addition of erlotinib to the gemcitabine/oxaliplatin regimen did not result in better PFS and OS outcomes than gemcitabine/oxaliplatin alone [31]. It is known that EGFR tyrosine kinase inhibitors (EGFR-TKIs) selectively target mutant EGFR better than wild-type EGFR; therefore, using EGFR-TKIs for CCA may not be a good option. In contrast, cetuximab is a monoclonal antibody that targets the wild-type EGFR; however, cetuximab also failed to demonstrate superior efficacy in a randomized phase II study [32,33], although a single-arm phase II study demonstrated promising results [34]. Interestingly, *RAS* mutation played no role in additional cetuximab treatment, which is a predictive factor for colorectal cancer. The addition of panitumumab, another monoclonal antibody targeting EGFR, to chemotherapy failed to show prolonged OS in a randomized phase II study of 89 patients with *KRAS* wild-type advanced BTC [35]. Further studies should focus on investigation-predictive biomarkers; otherwise, targeting EGFR may not be a possible strategy for CCA.

The use of trastuzumab, an antibody targeting HER2, in combination with chemotherapy showed a promising response in gallbladder cancer with *HER2*/*neu* genetic aberrations or protein overexpression, but no therapeutic effects on CCA in a small retrospective cohort [36]. Currently, there is an ongoing phase I study on the combined use of trastuzumab and tipifarnib (a farnesyltransferase inhibitor of RAS kinase) [37]. In addition, trastuzumab deruxtecan (DS-8201) is currently being studied in CCA patients with HER2 alterations [38]. DS-8201 has been approved for breast cancer with amplified/overexpressed HER2 [39] and has been widely studied in lung cancer, gastric cancer, and other solid cancer with HER2 alterations. Neratinib is an oral, irreversible, pan-HER tyrosine kinase inhibitor and is under investigation in a basket SUMMIT study (NCT01953926), which demonstrated that neratinib has encouraging clinical activity in multiple types of HER2-mutant solid tumor malignancies. Unfortunately, the study of BTC cohort did not meet its prespecified criteria for further expansion. The ORR was 16% and DCR was 28% among 25 BTC patients with HER-2 mutation. A subset of patients had tumor response and durable disease control, suggesting its antitumor activity in this rare population [40].

### 3.5. Vascular Endothelial Growth Factor Overexpression

Vascular endothelial growth factor (VEGFR) overexpression has been found in more than half of CCA patients, even though it has a low frequency of genetic mutations, which implies that this alteration is not the main driver gene [4,41,42,43]. Clinical trials of drugs such as bevacizumab, sorafenib, vandetanib, regorafenib, and ramucirumab, which target the signaling pathways in CCA, have yielded unsatisfactory results [4,44]. An early study of bevacizumab in combination with chemotherapy demonstrated a median PFS of 7 months, and PFS at 6 months was 63% [45]; however, no randomized studies have been conducted. Previous studies were conducted among unselected patients; therefore, predictive biomarkers, such as VEGFR overexpression, should be included in patient selection, which may improve the efficacy of bevacizumab in CCA patients.

Regorafenib, a multi-kinase inhibitor, showed modest efficacy compared to best supportive care after the failure of chemotherapy in REACHIN, a randomized, double-blind, phase II clinical trial [46]. In this study, median PFS (the primary endpoint) was modestly but significantly increased in regorafenib use (3 vs. 1.5 months, *p* = 0.004), and the DCR was also higher (70% vs. 33%); however, no improvement in OS was observed (median OS: 5.3 vs. 5.1 months, *p* = 0.28) compared to the best supportive care. This study was performed in unselective patients and was not a biomarker-driven study.

A phase II study (LEAP-005, NCT03797326) of anobinost in combination with pembrolizumab demonstrated an ORR of 16% and DCR of 58% in previously treated among 31 BTC patients [47].

### 3.6. KRAS Mutations

KRAS is the most frequently mutated isoform of RAS mutations and was considered an undruggable target before the development of sotorasib [48]. KRAS mutations commonly occur in CCA patients (20–25%), but there is no approved treatment for this [49]. Recently, sotorasib, a TKI that specifically targets KRAS^G12C^ mutation, has been approved for patients with previously treated non-small-cell lung cancer [50]. KRAS^G12C^ accounted for 2.3% of BTC in Chinese population [49]. Adagrasib is a KRAS^G12C^ inhibitor that irreversibly and selectively binds KRAS^G12C^ and demonstrated a good ORR of 41% among 27 patients with KRAS^G12C^-mutant and evaluable gastrointestinal tumors (including 8 BTC patients) in KRYSTAL-1 study [51]. Other KRAS^G12C^ inhibitors have also been studied in clinical trials.

In addition, by targeting alterations in the RAS/RAF/MEK/ERK signaling pathways, some studies have demonstrated the potential and efficacy of therapies targeting MEK, such as the MEK inhibitors selumetinib and trametinib, without selection of driver mutations. Selumetinib combined with chemotherapy has been studied [52]. At present, there is a phase II study (NCT02151084) in which the efficacy of selumetinib plus gemcitabine is compared with that of gemcitabine alone. 

Furthermore, autophagy deregulation was found to be associated with malignant cells compared with normal cholangiocytes and correlated with metastatic disease and poor prognosis in CCA [53]. The combination with trametinib and hydroxychloroquine (autophagy inhibitor) has been investigated in KRAS-mutant BTC (NCT04566133).

### 3.7. BRAF Mutation

With an incidence of only 3–5%, BRAF^V600E^ mutations predominantly occur in iCCA [4,54]. There have been case reports of successfully combined therapies involving the BRAF inhibitor vemurafenib or dabrafenib, with or without the MEK inhibitor, for the treatment of CCA [55]. A phase II study of dabrafenib and trametinib demonstrated comparable efficacy (objective response rate of 51%) in BRAF^V600E^ mutated biliary tract cancer (91% were iCCA) [54] as this combination in *BRAF^V600E^* mutated melanoma [56] and lung cancer [57,58]. There are several phase I and II studies ongoing investigating the use of BRAF-targeted therapy in CCA. Although BRAF-targeted therapy could treat *BRAF^V600E^* (class I) mutant iCCA, other BRAF mutations (class II and III) remain untargeted. Belvarafenib [59], a potent, selective RAF dimer (type II) inhibitor, has been studied in the TAPISTRY trial (NCT04589845) with patients with class II/III *BRAF* mutations.

### 3.8. Abnormal Activation of the PI3K/AKT/mTOR Signaling Pathway

*PI3KCA* and *PTEN* mutations result in an abnormal PI3K/AKT/mTOR signaling pathway. Currently, there are clinical trials on CCA therapies that separately target PI3K, AKT, and mTOR, such as the PI3K inhibitors copanlisib and BKM120, AKT inhibitor MK2206, and mTOR inhibitor everolimus [60]. Only alpelisib is approved for PIK3CA-mutated hormone receptor-positive advanced breast cancer [61]. The TAPISTRY trial also enrolled patients with *PI3KCA* double mutations treated with inavolisib, and AKT1/2/3 mutations treated with ipatasertib. Other inhibitors have also been studied in clinical trials.

### 3.9. Chromatin Remodeling-Associated Gene Mutations

Chromatin remodeling is a dynamic genetic mechanism in which chromatin modifications govern gene activity. Chromatin remodeling-associated gene mutations such as *ARID1A*, *ARID1B*, *ARID2*, *PBRM1*, *BAP1*, *SMARCA2*, *SMARCA4*, and *SMARCAD1* may be detected in patients with CCA [4]. The DNA methyltransferase inhibitors decitabine and azacitidine, as well as the histone deacetylase inhibitors vorinostat, valproic acid, romidepsin, and panobinostat are drugs used in targeted therapies for chromatin remodeling-associated gene mutations [62]. In addition, EZH2, PARP, and mTOR inhibitors may be potential treatments in preclinical studies [63]. This is a large unmet need, and no clinical studies have demonstrated the efficacy of such inhibitors.

Interestingly, chromatin-remodeling genes may promote immunotherapy resistance identified in more than 100 genes, including PBRM1, ARID2, and BRD7 genes, which encode a specific Switch/Sucrose Nonfermentable (SWI/SNF) chromatin remodeling complex leading to more sensitive to T cell-mediated killing [64]. Another study found that loss-of-function mutations in the PBRM1 gene were associated with clinical benefits in clear cell renal cell carcinoma treated with immune checkpoint inhibitors [65]. A promising ORR of 89% was found in nine patients who had metastatic pancreatic adenocarcinoma harboring SWI/SNF chromatin remodeling gene alterations [66]. Regarding CCA, protein loss of any tested SWI/SNF subunit was associated with unfavorable survival in iCCA but not eCCA [67]; however, the impact of SWI/SNF and immune checkpoint inhibitors in CCA is unknown.

### 3.10. High Microsatellite Instability and High Tumor Mutational Burden

Defects in DNA mismatch repair (dMMR) proteins and subsequent MSI-H result in the accumulation of mutation loads and neoantigens, which stimulate the antitumor immune response of the host; therefore, MSI-I is considered a good predictive marker for immune checkpoint inhibitors [68] approximately 3% of CCA cases had dMMR or MSI-H in one study [69]. CCA patients with the MSI-H phenotype can be treated with pembrolizumab, based on the results of the Keynote 158 study [70]. This study enrolled 22 CCA patients with 9 (41%) objective responses and the median duration of response ranged from 4.1 to >24.9 months. Two responders experienced a complete response.

In contrast, although pembrolizumab can be used for solid tumors with a high tumor mutational burden (TMB-H), no patients with CCA were enrolled in the trial [71]. Consequently, the efficacy of pembrolizumab in CCA with TMB-H remains uncertain. In addition, the appropriate threshold to define high TMB in CCA is unknown, as validation studies were conducted mainly in other cancers, and thresholds for TMB are likely to vary across tumor types.

### 3.11. Targeting TP53

TP53 mutations frequently occur in CCA; therefore, targeting p53 and associated proteins may be a potential development [72,73]. MDM2 and WIP1 inhibitors may be the options for wild-type p53 [71,72]. Eprenetapopt (APR-246) is active for *TP53*-mutant myelodysplastic syndromes and may extend its study to solid cancer in the future [74].

### 3.12. Targeting Tumor Microenvironment (TME)

TME is a hypoxic, acidic, and immune/inflammatory cell-enriched area surrounding tumors. TME can directly interact with the CCA epithelium to support tumor growth and proliferation by the interplay of extracellular ligands in TME [75]; therefore, targeting TME such as cancer-associated fibroblasts, endothelial cells, and extracellular matrix may be a potential therapeutic strategy for cancer treatment [76]; however, most studies are in a preclinical setting. In addition, CCA can be sub-grouped into four immune subtypes based on TMB, and the inflamed CCA subtype was found, which is potentially treatable by immune checkpoint inhibitors [77].

## 4. Future Challenges and Other Novel Treatments

It should be noted that the previously mentioned and approved drugs are indicated in later-line settings. Chemotherapy remains the main treatment option for patients with or without unknown genetic alterations [78]. The efficacy of first-line targeted therapy compared to that of standard chemotherapy is unknown. Before successful clinical studies are carried out, we do not suggest frontline targeted therapy in clinical settings unless the patient is unfit for chemotherapy.

In addition to the targeted therapies against well-known driver mutations, novel treatments have been developed in preclinical studies. Unless clinical efficacy has been demonstrated in patients with CCA, the use of novel targeted therapies should be carefully evaluated. NGS can help physicians identify candidates for targeted therapy or immunotherapy; however, NGS can only detect genetic alterations, rather than protein expression. AXL, a receptor tyrosine kinase, is a member of the TAM family that binds to its high-affinity ligand growth arrest-specific protein 6 (GAS6). The binding of GAS6/AXL leads to tumor growth, invasion, metastasis, immune regulation, and stem cell maintenance [79]. As a result, monoclonal antibodies and antibody-drug conjugates have been investigated in cancers with AXL expression. Such protein expression cannot be assessed using NGS, which limits the utility of NGS in clinical practice.

In addition, some targeted therapies such as cetuximab, bevacizumab, and regorafenib are not mutation-driven; therefore, if specific biomarkers can be identified as predictive or prognostic factors, future studies should be designed to identify patients who may benefit from novel treatment. Moreover, simply harboring a potentially targetable mutation does not guarantee that the patient would benefit from targeted CCA therapy. If targeted testing identifies a potentially actionable genetic abnormality for which a molecularly targeted treatment is available but has not yet been approved (for example, *HER2*, *BRAF*, *PI3KCA*, and *AKT*), these patients should be enrolled in clinical trials, if possible.

However, intratumoral heterogeneity (ITH) of driver mutations and other mutations is present in a variety of cancers, which may lead to resistance to cancer treatment [80,81,82]. The evolution of ITH during cancer treatment has a negative impact on immunotherapy and targeted therapy; therefore, potential strategies to improve therapeutic outcomes by directly targeting ITH are warranted [82,83]; however, outside laboratory research, clinically available NSG cannot evaluate ITH, and methods to target ITH have not been validated in clinical settings.

Recently, the TOPAZ-1 trial [84] showed that the addition of durvalumab, an immune checkpoint inhibitor targeting PD-L1, in combination with gemcitabine and cisplatin improved survival in treatment-naïve patients with CCA. As a result, immunotherapy has been proven to play a role in unselected patients with CCA. This was a breakthrough as no successful phase III trials for treatment-naïve CCA had been reported after the ABC-02 trial in 2010. This was also the first study to improve the median OS of patients with CCA to >1 year and represented the beginning of the immunotherapy era in CCA. In addition to immunotherapy and chemotherapy, future studies on the combination of immunotherapy and targeted therapy are warranted.

The last issue concerns performance status (PS), biliary drainage, and comorbidities. In this review, we discussed the efficacy of targeted therapy and immunotherapy; however, safety and possible adverse events should be considered when such treatments are administered to patients. Currently, the benefit of chemotherapy is limited to patients with good PS. In contrast, PS is not a contraindication for targeted therapy and immunotherapy, but it is a critical prognostic factor for patients with CCA. When selecting treatment, the underlying comorbidity should be considered based on the toxicities of the treatment. Biliary obstruction frequently occurs in patients with advanced CCA; therefore, adequate drainage should be performed; therefore, the use of targeted therapy or immunotherapy in patients with hyperbilirubinemia should be performed with caution.

## 5. Conclusions

Precision medicine has become mainstream, following the almost complete decoding of the human genome. Because of the high heterogeneity of CCA tumors, tumor microenvironmental factors, and the complexity of CCA molecular biology, pemigatinib, infigratinib, ivosidenib, larotrectinib, and entrectinib remain the only approved drugs for the treatment of CCA with *FGFR2* fusion, *IDH1* mutation, and *NTRK* fusion. Dabrafenib and trametinib have demonstrated great efficacy in CCA with *BRAF* mutations; however, this combination has not yet been approved by FDA. The development of breakthrough drugs for treating CCA includes FGFR2, PI3K/AKT/mTOR, and IDH inhibitors. Precision medicine for CCA can be enhanced by a better understanding of the genetic expression of CCA, developing innovative targeted therapies, and conducting personalized clinical trials on different CCA genotypes (Table 4). There is further need for improvement in precision medicine therapies in the treatment of CCA.

## Figures and Tables

**Figure 1 life-12-00829-f001:**
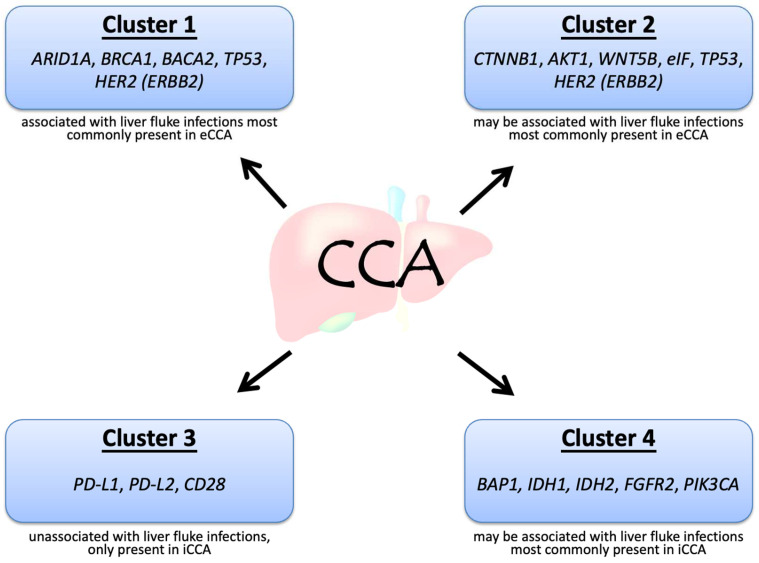
Molecular biological clusters of cholangiocarcinoma, adapted from Mahipal et al. [7].

**Figure 2 life-12-00829-f002:**
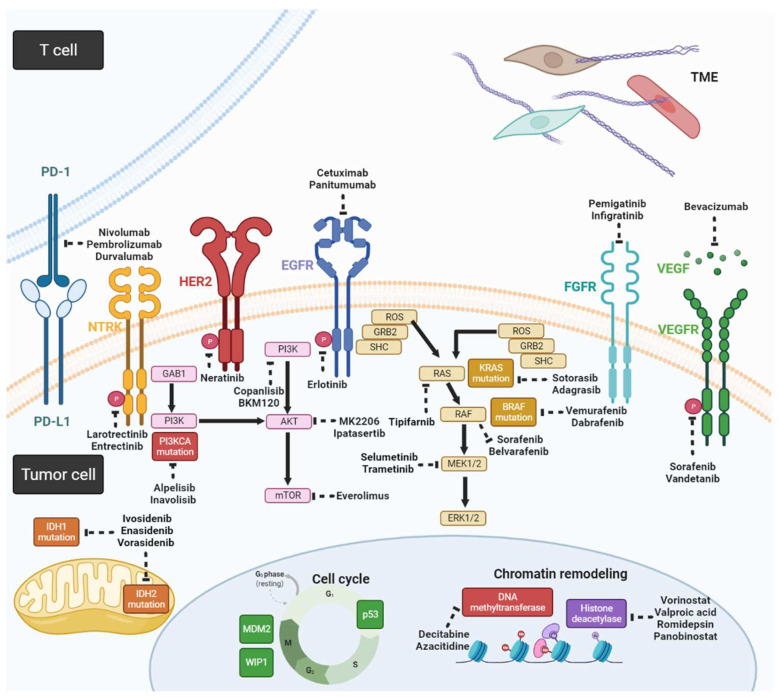
Schema for potential targeted agents and related signaling pathways in CCA (created with https://biorender.com/ (accessed on 25 May 2022)).

**Table 1 life-12-00829-t001:** Acronyms and full names of genes involved in cholangiocarcinoma.

*ARID1A*	AT-rich interaction domain 1A
*AKT1*	Protein Kinase B
*BAP1*	BRCA associated protein 1
*BRAF*	v-raf murine sarcoma viral oncogene homolog B1
*BRCA1/2*	Breast cancer gene 1/2
*CDKN2A*	Cyclin-dependent kinase inhibitor 2A
*CTNNB1*	Catenin beta 1
*EGFR*	Epidermal growth factor receptor
*eIF*	Eukaryotic initiation factor
*FGFR2*	Fibroblast growth factor receptor 2
*HER2*	Human epidermal growth factor receptor 2
*IDH1/2*	Isocitrate dehydrogenase ½
*JAK/STAT*	Janus Kinase/Signal transducer and activator of transcription
*KRAS*	Kirsten rat sarcoma viral oncogene homolog
*MEK*	Mitogen-activated protein kinase
*MET*	MET proto-oncogene, receptor tyrosine kinase
*mTOR*	Mammalian target of rapamycin
*NRAS*	Neuroblastoma ras viral oncogene homolog
*NTRK*	Neurotrophic tyrosine receptor kinase
*PBRM1*	Poly-bromo1
*PIK3CA*	Phosphatidylinotitol 3-kinase catalytic subunit alpha
*PTEN*	Phosphatase and tensin homolog
*ROS1*	ROS proto-oncogene 1
*TP53*	Tumor protein 53
*VEGF*	Vascular endothelial growth factor
*WNT5B*	Wnt family 5B gene

**Table 2 life-12-00829-t002:** Potential candidates in cholangiocarcinoma treatment by target gene alterations.

Genetic Alteration	Targeted Therapies
*ARID1A*	HDAC inhibitors, EZH2 inhibitors, PARP inhibitors, mTOR inhibitors
*BAP1*	HDAC inhibitors, EZH2 inhibitors, PARP inhibitors
*BRAF*	Dabrafenib, Vemurafenib, Trametinib, Selumetinib
*CDKN2A*	CDK4/6 inhibitors
*EGFR*	Cetuximab
*FGFR2*	Infigratinib, Derazantinib, Erdafitinib, Futibatinib, Pemigatinib, Ponatinib, Debio 1347, FRA144, INCB054828, NVP-BGJ398, INCB054828
*HER2*	Trastuzumab
*IDH1*	Ivosidenib (AG-120), Vorasidenib (AG-881)
*IDH2*	Enasidenib (AG-221), Vorasidenib (AG-881)
*JAK/STAT*	Tofacitinib, Baricitinib, Peficitinib, Upadacitinib, Filgotinib
*KRAS*	MEK inhibitors: Trametinib, Selumetinib
*MET*	Capmatinib, Tepotinib
*NRAS*	MEK inhibitors: Trametinib, Selumetinib
*NTRK*	larotrectinib or entrectinib
*PBRM1*	PARP inhibitors, Immume checkpoint inhibitors
*PIK3CA*	PIK3CA/AKT/mTOR inhibitors, Copanlisib, BKM12, MK2206, everolimus
*PTEN*	PIK3CA/AKT/mTOR inhibitors
*ROS1*	Crizotinib, Ceritinib
*TP53*	Wee1 inhibitors: Adavosertib (AZD1775), MDM2 inhibitors: idasanutin
*VEGF*	Bevacizumab, Sorafenib, Vandetanib, Regorafenib, Ramucirumab

**Table 3 life-12-00829-t003:** Outcomes of clinical trials of small-molecule targeted therapies for fibroblast growth factor receptor (FGFR) pathways.

Targeted Therapy	FGFR	N	CR/PR (%)	SD (%)	DCR (%)	PFS (Months)
Pemigatinib	FGFR 1–3	107	35.5	46.7	82.5	6.9
Infigratinib	FGFR 1–3	108	23.1	NR	NR	7.3
Derazantinib	FGFR 1–3	29	20.7	62.1	82.8	5.7
Erdafitinib	FGFR 1–4	11	27	27	55	5.1
Futibatinib	FGFR 1–4	45	25	53	79	NR

**Table 4 life-12-00829-t004:** Selected ongoing clinical trials in cholangiocarinoma (CCA), created by https://clinicaltrials.gov/ (accessed on 20 May 2022).

Type of CCA	Phase	Selected Patients	Drugs	ClinicalTrials.gov Identifier
BTC	II	All	Gemcitabine + Cisplatin + Selumetinib (MEK)	NCT02151084
BTC	II	All	Pembrolizumab + Lenvatinib (VEGF)	NCT03797326 (LEAP-005)
iCCA	III	FGFR2 fusion	Futibatinib vs. Gemcitabine + Cisplatin	NCT04093362 (FOENIX-CCA3)
iCCA	II	FGFR1-3 fusion	Debio-1347	NCT03834220 (FUZE)
iCCA	II	IDH1 mutation	Ivosidenib + nivolumab	NCT04056910
iCCA	I/II	IDH1 mutation	Enasidenib (AG-221)	NCT02273739
BTC	II	RAS mutation	Trametinib + Hydroxychloroquine	NCT04566133
BTC (iCCA)	II	BRAF mutations (class II/III)	Belvarafenib	NCT04589845 (TAPISTRY)
BTC (eCCA)	II	HER2 mutation	Neratinib	NCT01953926 (SUMMIT)

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
