# Peer review of "Precision Medicine in Cholangiocarcinoma: Past, Present, and Future"

_life, 2022, doi:10.3390/life12060829_

Round 1
Reviewer 1 Report
The authors present a review on precision medicine in cholangiocarcinoma based on genetic profiling.The review is comprehensive and gives clear picture of the state of the field and how we got there.
Some recomendations would improve the quality and interest in readers:
- Genetic alterations and signalling pathways in CCA could be incorporated in a schematic figure for a better understanding of the cellular processes involved in target therapies.
- It has long been known that the response to target therapies does not only depend on the success on tumor cells. What is the impact of these signaling pathways on the tumor microenvironment? A section relating these aspects with other components of the tumor microenvironment could add greater impact.
Author Response
We thank the Reviewer for these positive and helpful suggestions regarding our manuscript.
1. Genetic alterations and signalling pathways in CCA could be incorporated in a schematic figure for a better understanding of the cellular processes involved in target therapies.
Reply: I have added a schematic figure in the revised manuscript.
2. It has long been known that the response to target therapies does not only depend on the success on tumor cells. What is the impact of these signaling pathways on the tumor microenvironment? A section relating these aspects with other components of the tumor microenvironment could add greater impact.
Reply: Thanks for the suggestion. We have already add the microenvironment at section 3.12 in this manuscript.
Reviewer 2 Report
Dear authors, congratulations on your conduct for review titled "Precision Medicine in Cholangiocarcinoma: Past, Present, and Future". Please find my suggestions below:
Minor issues:
Please use an online tool to check for plagiarism. Some sentences seem to be copy/pasted from the original reference.
First page, lines 11-13 and 41-42: "The median survival time...is only 10%"; would recommend using this statement only once and not repeating it in both both abstract and Introduction
Page 2, lin 63: "mutations in IDH1, IDH2, FGFR2, BAP1....". Please consider changing as FGFR2 are mostly fusions/rearrangements. Can consider changing to "Alterations in IDH1, ..." or "Mutations in IDH1, IDH2, BAP1 and rearrangements in FGFR2 are often present ..."
Figure 1: cluster 4: "most commonly present in eCCA" is incorrect. Please change to "most commonly present in iCCA"
Table 2: PBRM1: "Protein Poly-bromo1" can potentially be changed to "Poly-bromo1"
Table 2: PIK3CA: please change "Phosphatidylinotitol 3-kinase CA" to "Phosphatidylinotitol 3-kinase catalytic subunit alpha"
Table 2: ROS1: please change "c-ros oncogene 1" to "ROS proto-oncogene 1"
Page 4, line 90: Consider changing "amplification, fusion" to "amplifications and fusions". Also, "leading to gain of function mutations, which may": considering change in "leading to gain of function, which may..."
Page 4, line 100: please change "FGFR2 functions" to "FGFR2 fusions".
Page 4, line 101: please note infigratinib is also FDA approved.
Page 4, line 140: EGFR overexpression is common in Biliary Tract Cancers (BTCs). Consider revising this sentence or deleting it.
Page 4, line 165: consider including data from SUMMIT basket trial and use of Neratinib in biliary tract cancer.
Page 6, section on VEGF: please consider adding data on Lenvatinib and Pembrolizumab in Biliary tract cancer. "https://oncologypro.esmo.org/meeting-resources/esmo-virtual-congress-2020/leap-005-phase-ii-study-of-lenvatinib-len-plus-pembrolizumab-pembro-in-patients-pts-with-previously-treated-advanced-solid-tumours"
Page 6, KRAS mutation: Please consider re-organizing and re-writing. Consider starting with the last sentence (lines 194-195 currently). You can follow that by the prevalence of KRAS G12C in CCA followed by the result of the KRYSTAL-1 study presented in GI ASCO 2022, showing that among the 8 patients with biliary tract cancer, 4 had response (50% response rate). NTC04566133. Then can consider moving to downstream inhibition. Apart from what you have involved, please consider involving data on rationale and value of MEK inhibition plus autophagy inhibition (such as Trametinib/Hydroxychloroquine); Refer to the active clinical trial of this combination in biliary tract cancer.
Page 6, section 3.9: please refer to the potential role of such alterations in immunotherapy with check-point inhibition.
Table 4: Seems very similar to table 3 of reference #6. If you used that as your reference, please refer to it in the body of the manuscript.
Author Response
We thank the Reviewer for these positive and helpful suggestions regarding our manuscript.
Minor issues:
Please use an online tool to check for plagiarism. Some sentences seem to be copy/pasted from the original reference.
Reply: We have checked or plagiarism before submission.
First page, lines 11-13 and 41-42: "The median survival time...is only 10%"; would recommend using this statement only once and not repeating it in both both abstract and Introduction
Reply: We have revised the statement in the introduction of revised manuscript.
Page 2, lin 63: "mutations in IDH1, IDH2, FGFR2, BAP1....". Please consider changing as FGFR2 are mostly fusions/rearrangements. Can consider changing to "Alterations in IDH1, ..." or "Mutations in IDH1, IDH2, BAP1 and rearrangements in FGFR2 are often present ..."
Reply: Thanks for your comment. Generally, a fusion/rearrangement is considered one type of mutation. To avoid misunderstanding and use exact phrases, we have changed this based on your suggestion in the revised manuscript.
Figure 1: cluster 4: "most commonly present in eCCA" is incorrect. Please change to "most commonly present in iCCA"
Reply: Thanks for your comment. We have changed figure 1 in the revised manuscript.
Table 2: PBRM1: "Protein Poly-bromo1" can potentially be changed to "Poly-bromo1"
Reply: Thanks for your comment. We have revised this in the revised manuscript.
Table 2: PIK3CA: please change "Phosphatidylinotitol 3-kinase CA" to "Phosphatidylinotitol 3-kinase catalytic subunit alpha"
Reply: Thanks for your comment. We have revised this in the revised manuscript.
Table 2: ROS1: please change "c-ros oncogene 1" to "ROS proto-oncogene 1"
Reply: Thanks for your comment. We have revised this in the revised manuscript.
Page 4, line 90: Consider changing "amplification, fusion" to "amplifications and fusions". Also, "leading to gain of function mutations, which may": considering change in "leading to gain of function, which may..."
Reply: Thanks for your comment. We have revised this in the revised manuscript.
Page 4, line 100: please change "FGFR2 functions" to "FGFR2 fusions".
Reply: Thanks for your comment. We have revised this in the revised manuscript.
Page 4, line 101: please note infigratinib is also FDA approved.
Reply: Thanks for your comment. We have revised this in the revised manuscript.
Page 4, line 140: EGFR overexpression is common in Biliary Tract Cancers (BTCs). Consider revising this sentence or deleting it.
Reply: Thanks for your suggestion. We have deleted the sentence.
Page 4, line 165: consider including data from SUMMIT basket trial and use of Neratinib in biliary tract cancer.
Reply: Thanks for your comment. We have added this in the revised manuscript.
Page 6, section on VEGF: please consider adding data on Lenvatinib and Pembrolizumab in Biliary tract cancer. https://oncologypro.esmo.org/meeting-resources/esmo-virtual-congress-2020/leap-005-phase-ii-study-of-lenvatinib-len-plus-pembrolizumab-pembro-in-patients-pts-with-previously-treated-advanced-solid-tumours
Reply: Thanks for your comment. We have added this in the revised manuscript.
Page 6, KRAS mutation: Please consider re-organizing and re-writing. Consider starting with the last sentence (lines 194-195 currently). You can follow that by the prevalence of KRAS G12C in CCA followed by the result of the KRYSTAL-1 study presented in GI ASCO 2022, showing that among the 8 patients with biliary tract cancer, 4 had response (50% response rate). NTC04566133. Then can consider moving to downstream inhibition. Apart from what you have involved, please consider involving data on rationale and value of MEK inhibition plus autophagy inhibition (such as Trametinib/Hydroxychloroquine); Refer to the active clinical trial of this combination in biliary tract cancer.
Reply: Thanks for your comments. We have revised this paragraph of revised manuscript.
Page 6, section 3.9: please refer to the potential role of such alterations in immunotherapy with check-point inhibition.
Reply: Thanks for your comments. We have revised this section of revised manuscript.
Table 4: Seems very similar to table 3 of reference #6. If you used that as your reference, please refer to it in the body of the manuscript.
Reply: Thanks for your comments. All the data was collected from previous reports and summarized in a Table 3. We have done revision for this table.
Reviewer 3 Report
The review written by Cheng and colleagues concerns the new personalized medicine approaches currently under investigation for the treatment of cholangiocarcinoma (CCA) in its various sub-variants (intrahepatic, extrahepatic, ...). CCA is an extremely aggressive and deadly tumor for which classical chemotherapy treatments are largely ineffective and for which little progress has been made regarding treatment were performed in the last years. The review examines recent literature data that identify a series of mutations that can affect the biliary tree at the origin of CCA. The majority of these genetic variants and the pathways that they supervise, are now potentially targetable. In general, the review is easy to follow, although there are some typos to be corrected.
The major concerns are:
- I would add a table summarizing the main features of the most promising clinical trials actually under investigation (clinical trial code number, type of CCA, drugs or inhibitors used, phase,…).
- Given the increasing importance of the tumor microenvironment (TME) in the pathogenesis of CCA, as demonstrated by some recent studies and reviews, I would add a brief chapter concerning the target therapy for TME.
Minor Concerns:
Line 33: among the best-known and studied causes of CCA, I would add Caroli's disease and, in general, fibropolycystic liver diseases.
Author Response
We thank the Reviewer for these positive and helpful suggestions regarding our manuscript.
The major concerns are:
- I would add a table summarizing the main features of the most promising clinical trials actually under investigation (clinical trial code number, type of CCA, drugs or inhibitors used, phase,…).
Reply: Thanks for your comments. We have added this in the revised manuscript (Table 4).
- Given the increasing importance of the tumor microenvironment (TME) in the pathogenesis of CCA, as demonstrated by some recent studies and reviews, I would add a brief chapter concerning the target therapy for TME.
Reply: Thanks for your comments. We have added this in the revised manuscript (section 3.12).
Minor Concerns:
Line 33: among the best-known and studied causes of CCA, I would add Caroli's disease and, in general, fibropolycystic liver diseases.
Reply: Thanks for your comments. We have revised this in the revised manuscript.
Round 2
Reviewer 1 Report
the authors have considered my appreciations and incorporated changes in the manuscript that increase its quality. I recommend its approval.
Reviewer 3 Report
The review is actually improved and worthy to be published.